# Intracellular Water Content in Lean Mass is Associated with Muscle Strength, Functional Capacity, and Frailty in Community-Dwelling Elderly Individuals. A Cross-Sectional Study

**DOI:** 10.3390/nu11030661

**Published:** 2019-03-19

**Authors:** Mateu Serra-Prat, Isabel Lorenzo, Elisabet Palomera, Juan Carlos Yébenes, Lluís Campins, Mateu Cabré

**Affiliations:** 1Research Unit, Consorci Sanitari del Maresme, 08304 Mataró (Barcelona), Spain; ilorenzo@csdm.cat (I.L.); epalomera@csdm.cat (E.P.); 2Intensive Care Unit, Consorci Sanitari del Maresme, 08304 Mataró (Barcelona), Spain; jyebenes@csdm.cat; 3Pharmacy Department. Consorci Sanitari del Maresme, 08304 Mataró (Barcelona), Spain; lcampins@csdm.cat; 4Geriatric Department, Consorci Sanitari del Maresme, 08304 Mataró (Barcelona), Spain; mcabre@csdm.cat

**Keywords:** cell hydration, intracellular water, aging, muscle strength, frailty, functional capacity

## Abstract

High intracellular water (ICW) content has been associated with better functional performance and a lower frailty risk in elderly people. However, it is not clear if the protective effect of high ICW is due to greater muscle mass or better muscle quality and cell hydration. We aimed to assess the relationship between ICW content in lean mass (LM) and muscle strength, functional performance, frailty, and other clinical characteristics in elderly people. In an observational cross-sectional study of community-dwelling subjects aged ≥75 years, ICW and LM were estimated by bioelectrical impedance, and the ICW/LM ratio (mL/kg) calculated. Muscle strength was measured as hand grip, frailty status was assessed according to Fried criteria, and functional status was assessed by Barthel score. For 324 recruited subjects (mean age 80 years), mean (SD) ICW/LM ratio was 408 (29.3) mL/kg. The ICW/LM ratio was negatively correlated with age (r_s_ = −0.249; *p* < 0.001). A higher ICW/LM ratio was associated with greater muscle strength, better functional capacity, and a lower frailty risk, even when adjusted by age, sex, nº of co-morbidities, and LM. ICW content in LM (including the muscle) may influence muscle strength, functional capacity and frailty. However, further studies are needed to confirm this hypothesis.

## 1. Introduction

Water is an essential nutrient with structural, metabolic, transport and temperature control functions in the body. Total body water (TBW), which represents approximately 55–60% of the body weight of adults, is distributed in intracellular water (ICW) and extracellular water (ECW) compartments, with the latter including plasma and interstitial fluid. Water exchange between intra- and extracellular compartments is regulated mainly by osmotic pressure, with water flowing across the cell membrane to balance osmolarity on each side [1]. This transport is mediated by a family of membrane protein channels, called aquaporins (AQPs), expressed in response to specific stimuli [2]. Active transport of osmotically active compounds across the cell membrane indirectly contributes to water diffusion that balances ICW and ECW osmolarities. ECW osmolarity needs to remain within very narrow limits (285–295 mOsm/kg) despite fluctuations in water and solute intake. Increased ECW osmolarity involves a flow of water out of the cell, leading to cell shrinkage and important structural and functional protein alterations [3]. When cells are dehydrated, enzyme activity is altered, and the damaging effects on the cytoskeleton and the nucleus [4] lead to apoptosis and cell death. Osmotic homeostasis is mainly regulated by arginine vasopressin (AVP), stimulated by osmoreceptors and baroreceptors and released to the bloodstream by the posterior pituitary. AVP stimulates water reabsorption and urine concentration through AQP2 expression in the collecting duct [5]. An impaired kidney function reduces the ability to concentrate urine and to maintain plasmatic osmolarity within the physiological range. Elevated extracellular osmolarity directly stimulates tumor necrosis factor (TNF) and interleukin (IL) 1, 6, and 8 secretion, contributing to varying disorders related to a strong inflammatory response, such as chronic arthritis and inflammatory bowel diseases [6], and also to liver disorders, insulin resistance, diabetes, hypertension, cardiovascular diseases, and carcinogenesis [3,7]. Despite the evidence, however, the consequences of hyperosmotic stress for aging processes have been little studied and are not well understood.

Muscle is the most extensive organ in the body, representing approximately 40% of total body weight and 50% of body lean mass (LM). Muscle function is associated with successful aging and reflects the integrated health status in aged population [8]. Since 76% of muscle content is water, water loss can be expected to affect muscle function. TBW decreases with age mainly because of a reduction in LM and muscle mass relative to fat mass [9]. However, a decrease in ICW with age has also been reported [10], which may be related to the elevated plasma osmolarity observed in aged individuals. This plasma hyperosmolarity is largely due to the reduced ability of elderly adults to concentrate urine [11], which, combined with elevated levels of AVP, point to a reduced renal sensitivity to AVP in elderly individuals [5]. A recent publication of our group shows that elderly people with higher ICW content had better functional performance and a lower frailty risk, suggesting the protective effects of cell hydration [12]. However, it was not clear from that study whether the observed positive effect of high ICW content was due to greater muscle mass or to better muscle quality and cell hydration. 

The objective of this study was to explore the possible links between the ICW/LM ratio (as an indicator of muscle quality and cell hydration) and muscle strength, functional performance, frailty, and other clinical characteristics in elderly people.

## 2. Materials and Methods

### 2.1. Study Design and Population

An observational cross-sectional study was designed, for a sample randomly selected from a population of community-dwelling subjects aged 75 years and older in the catchment area of 3 primary care centers in the city of Mataró (Barcelona, Spain) during the period January–July 2014. Subjects were excluded if they were institutionalized, had active malignancy, dementia or serious mental illness, or had a life expectancy of under 6 months. Details of the study methodology have been previously reported [13]. The institutional research ethics committee approved the study protocol (code CEIC CSdM 64/13) and all participants gave their written informed consent before inclusion.

### 2.2. Study Factors and Data Collection

The ICW content in LM (ICW/LM ratio, expressed in mL/kg) was used as an indicator of muscle quality and cell hydration. ICW and LM were assessed by bioelectrical impedance analysis (BIA, Pontassieve, Italy) (EFG3 Bioelectrical Impedance Analyser Electrofluidgraph, Akern SRL), a non-invasive, reproducible and validated method of analysis that—on the basis of the body’s ability to conduct electrical current—accurately calculates fat mass, LM, and muscle mass (in kg and as a percentage of TBW), and ECW and ICW (in liters and as a percentage of TBW) [14]. The method functions by measuring the opposition of biological tissues to the passage of an alternating low-intensity electrical current (impedance). Two vectors are considered: resistance (R), which is the opposition to the flow of electrons, inversely proportional to TBW; and reactance (Xc), which is the opposition of cell membranes to the passage of electric current (reflecting cell membrane integrity and LM). BIA was performed after ensuring that patients did not show evident clinical signs or symptoms of dehydration (hypotension, tachycardia, sign of the skin fold, dry mouth, dry eyes, decreased production of urine, or dizziness) and following the standard instructions for the correct execution of the BIA, which include hydration and anthropometric assessment conditions and test performance conditions (such as removal of metal objects or extremities separated from the body). Muscle strength was estimated from hand-grip strength, calculated (in kg) using a hand-held JAMAR dynamometer. Three measures of hand grip were performed in the dominant hand, and the maximum value of these three was used in the analysis. Frailty status was assessed according to Fried criteria [15] in terms of weight loss, exhaustion, poor physical activity, low gait speed, and weakness; thus, a patient was considered robust when no frailty criteria were met, pre-frail when 1 or 2 criteria were met and frail when 3 or more criteria were met. Functional capacity was assessed by the Barthel score, the timed up-and-go test (TUG), gait speed, outdoor life (if the patient leaves home) and outdoor walking (number of hours a day walking out of home). Other study variables included socio-demographic characteristics, co-morbidities, chronic medication and nutritional status assessed using the short-form Mini Nutritional Assessment (MNA_sf). Information on co-morbidities and medication was obtained from patient medical records and all other information was obtained directly from the patient by trained healthcare professionals.

### 2.3. Statistical Analysis

Spearman’s correlation coefficient (r_s_) was used to assess correlations between the ICW/LM ratio and other continuous variables such as muscle strength, Barthel score and gait speed. For dichotomous variables such as sex or frailty, the mean ICW/LM ratios between categories were compared using the t-test or the Mann–Whitney *U* test, depending on the normal distribution of the ICW/LM ratio in the analyzed groups. Simple linear regression analyses were performed to assess the relationship between the ICW/LM ratio and the other study variables and beta (β) coefficients were calculated. Moreover, multiple linear regression analysis was used to assess the independent effect of ICW/LM ratio on hand grip strength and Barthel score, and simple and multiple logistic regression analyses were performed to estimate the crude and adjusted effect of the ICW/LM ratio on frailty. Variables included in multivariate models were those associated with ICW/LM ratio in the bivariate analysis (age, sex, number of co-morbidities, and LN). Medications were not included in the multivariate analyses because of multi-collinearity with number of co-morbidities. Statistical significance was set to *p* < 0.05 for all analyses. The program IBM SPSS 15.0 was used for the statistical analysis (SPSS Inc., Chicago, IL, USA).

## 3. Results

### 3.1. Description of Sample Characteristics

Recruited were 324 individuals (170 men and 154 women), with a mean (SD) age of 80 (3.5) years. Table 1 describes the main clinical characteristics of the study sample. In the overall sample, the mean (SD) ICW/LM ratio was 408 (29) mL/kg (min-max 318–516 mL/kg), and values for 95% of the sample were between 344 mL/kg (percentile 2.5) and 468 mL/kg (percentile 97.5). By sex, the mean (SD) ICW/LM ratio for men was 417 (27) mL/kg (ranging from 357 mL/kg to 471 mL/kg for 95% of the male sample) and for women was 398 (29) mL/kg (ranging from 341 mL/kg to 470 mL/kg for 95% of the female sample). Sex differences for the ICW/LM ratio were statistically significant (*p* < 0.001). The ICW/LM ratio was negatively correlated with age, with r_s_ = −0.249 (*p* < 0.001) and a linear regression coefficient β = −2.07 (*p* < 0.001). The ICW/LM ratiowas associated with the number of co-morbidities (r_s_ = −0.117; *p* =< 0.001 and β = −1.43; *p* = 0.112) and also with the number of medications (r_s_ = −0.251; *p* =< 0.001 and β = −2.24; *p* < 0.001).

### 3.2. ICW/LM Ratio Relationships with Muscle Strength, Functional Capacity, and Frailty

Table 2 and Table 3 show the relationship between the ICW/LM ratio and functional capacity indicators, frailty criteria and frailty status. The ICW/LM ratio mean (SD) for robust persons was 414 mL/kg (28), for pre-frail persons 409 (29), and for frail persons 391 (26) (*p* < 0.001).All the studied functional capacity parameters were consistently associated with the ICW/LM ratio; thus, higher ICW/LM ratio was related to greater strength, better functional capacity and a lower frailty risk.

Table 4, which shows the relationship between the ICW/LM ratio and hand grip, Barthel score and frailty after adjustments for age, sex, number of co-morbidities and LM, confirms an independent effect of ICW/LM ratio on muscle strength, functional capacity and frailty.

The relationships between LN and indicators of muscle strength, functional capacity and frailty are shown as Appendix A.

## 4. Discussion

Our main results indicate that the ICW/LM ratio decreases with age in elderly people, is higher in older men than in older women, and that a higher ICW/LM ratio is independently associated with greater muscle strength, better functional capacity and a lower frailty risk. We propose the ICW/LM ratio as an indicator for intracellular space size in LM, which may reflect cell hydration or extracellular space expansion, both of which indicate LM quality and, indirectly, muscle mass quality. As expected, the ICW/LM ratio varied relatively little, oscillating as it did between fairly narrow values, and it was associated in a plausible and coherent way with some clinical characteristics such as age, sex or the number of co-morbidities. Therefore, we think that it could be a useful indicator of muscle quality in clinical research, although more studies are needed to confirm it.

Regarding the observed negative correlation between ICW/LM ratio and age, our results are consistent with well-established evidence indicating a loss in TBW and an increased risk of dehydration with age [16]. The risk of dehydration is a multifactorial process influenced by several causes related with poor water intake [17] and water losses. It is known that the capacity to concentrate urine decreases with age [11], not only due to a reduced glomerular filtration rate but also due to an altered capacity of the aged kidney to adequately respond to AVP stimuli [2,4,5]. This altered capacity results in an increase in plasmatic osmolarity, which promotes water flow to the extracellular compartment and cell shrinkage. Hyperosmotic stress has been associated with metabolic and cardiovascular diseases, which are probably related to the AVP response that also stimulates adrenocorticotropic hormone and cortisol release [2,7,18], and with chronic kidney disease, and a heightened inflammatory response [6], which play a role in carcinogenesis [19], muscle wasting, and aging [20]. The fact that, in our study, men had a higher ICW/LM ratio than women may be partially explained by the observation that men also had higher AVP plasma concentrations and more pronounced AVP-mediated effects on renal and vascular targets [5].

The main results of our study point to a robust relationship between ICW/LM ratio and hand grip, different indicators of functional capacity (especially Barthel and TUG scores), and frailty—even in the multivariate analysis after adjusting for age, sex, co-morbidities, and LM. The ICW/LM ratio showed an independent association with frailty with an adjusted OR = 0.98. In our opinion, this statistically significant effect is also of clinical relevance, as it indicates an approximately 2% reduction in the risk of frailty for each mL of ICW/kg LM increase. Thus, an increase of 1 SD of ICW/LM ratio (29 mL/kg) would correspond to an approximately 50% reduction in the risk of frailty. The independent effect of ICW/LM ratio on muscle strength is also statistically significant but of more moderate clinical relevance, with an increase of approximately 0.8kg of hand grip strength for 1 SD increases in ICW/LM ratio. We want to emphasize that the effect of ICW/LM ratio on muscle strength, functionality and frailty is independent of LM, indicating the importance of intracellular hydration in muscle functioning. These results, suggesting a relevant effect of low-grade chronic cell dehydration on muscle function, corroborate Yamada’s findings of a relative reduction in ICW compared to ECW in skeletal muscle in aging individuals [10], and also of a high ECW/ICW ratio as an independent predictor of poor muscle strength and gait speed in the elderly [21,22]. Muscle mass not only contains muscle cells but also an extracellular space containing ECW, fibrosis and infiltrated fat. The reduction in muscle cell mass and expansion in the extracellular space that appear with age may determine reduced muscle quality, poor muscle strength and impaired functional performance. Age-related loss of muscle strength is only partially explained by age-related loss of muscle mass [23]. That is why Clark et al proposed the term dynapenia to describe age-related loss of muscle strength and power and differentiate it from the sarcopenia concept, which, in the opinion of these authors, must be restricted to age-related loss of muscle mass [24]. Other factors related with muscle quality instead of muscle quantity must explain muscle weakness in elderly subjects. In our opinion, among these factors, myocite hydration could play a relevant role. Our results also agree with those published by Goulet et al., who suggest that acute dehydration of 1% of body weight in healthy active 60 to 75 years old mencould affect muscle endurance, power and strength [25]. Moreover, the correlation between ICW and muscle strength has also been observed in young adult athletes [26], with some authors suggesting that cell volume is a metabolic signal that regulates cellular function and also that cellular swelling leads to anabolism and cellular shrinkage promotes catabolism and protein degradation [27,28]. Although dehydration and hyperosmotic stress affect all bodily cells and tissues, these do not all respond in the same way. Nose et al. [29] assessed water loss as distributed among different compartments and different organs in rats that had been thermally dehydrated by 10% of their body weight, finding that, of the TBW loss, 59% was ECW loss—a proportion higher than the usual water distribution in the body (ECW represents approximately one third of TBW). The same authors [29], in reporting that 40% of TBW loss came from muscle, 30% from skin, 14% from bone, and 14% from viscera, suggested that muscle and skin play a major role in compensating for water loss and maintaining blood volume and circulation to the brain and liver.

On the other hand, Grazi has studied the interaction between water and muscle contractile proteins. The muscle cell contains thousands of myofibrils, formed by thin filaments (actin) and thick filaments (myosin), which interact to produce the shortening or contraction of the muscle. According to Grazi, the participation of water in muscle contraction is due to the hydrophilic nature of the proteins of the contractile apparatus and the non-ideal chemical property of such water solution [30]. This property implies that a small change in the concentration of these proteins induces a significant change in the osmotic pressure, as well as in the chemical potential of these proteins, which is related to the stiffness of the contractile structure (ability to oppose distension). Additionally, in vitro studies show that when the proteins of the contractile apparatus are well-hydrated they assemble to form ordered structures [30,31]. These authors describe that in the actin–myosin binding, the osmotic pressure determines both the distance between filaments and the elastic force that acts on the structure of each protein, so that this system links the osmotic pressure with the elastic reaction of the components of the cross-bridge. Overall, these studies have related water with muscle contraction and indicate that water determines sarcomere length, sarcomere stretching, and cross-bridge attachment and detachment [30,31]. It is also known that chronic kidney disease (CKD) results in muscle wasting [32]. CKD favors protein degradation because the activation of the ubiquitin–proteasome system, which can be initiated by complications of CKD, such as metabolic acidosis, defective insulin signaling, inflammation or impaired micro-RNA responses [33]. CKD is also related with impaired water homeostasis and hyperosmotic stress. Despite this, the mechanisms by which hyperosmotic stress and cell dehydration affects muscle strength have been little studied, and it is unknown to what extent the alteration of muscle function is due to a direct effect of cellular dehydration on actin–myosin interaction as suggested by Grazi [30] or is due to a process of protein degradation related with inflammation and favored by alterations in renal function. Further studies are still needed to confirm the role of intracellular dehydration in muscle function and functional capacity in the elderly. Our results suggest that ICW/LM ratio, a relatively easy and fast to obtain parameter in the consultation, may be useful as indicator of dehydration, frailty and poor functional capacity. Hyperosmolar dehydration is a frequent condition in routine clinical practice and in hospitalized older patients and is associated with an increased risk of death [34]. Thus, the ICW/LM ratio could be use fulin clinical practice as a comprehensive hydration assessment tool. However, its prognostic value requires further assessment by prospective studies.

This study has two main limitations. First, the cross-sectional design does not allow causal relationships to be established, as the directionality of the relationships is uncertain. Second, instructions for the correct realization of BIA were established, which include no intense physical exercise 24h before the test, no alcohol consumption 8h before the test, no water or fluid intake 2 h before the test or go to the toiled just before the test. Patients received these instructions when they were cited to the assessment visit, but the full compliance of these patient’s instructions cannot be guaranteed in all cases, which can influence BIA results. It remains to comment that, although BIA is not the gold standard for evaluating body composition, it is a validated, widely used, and well-accepted method for calculating accurate water distribution and body mass indicators (specifically, ECW, ICW, and LM).

In summary, the ICW/LM ratio obtained by BIA points to a robust relationship with age, sex, number of co-morbidities, and number of chronic medications, as well as an independent association with muscle strength, functional capacity, and frailty. These results suggest an important role in muscle function and frailty onset for ICW content in LM. However, further research is needed to corroborate these preliminary findings. 

## Figures and Tables

**Table 1 nutrients-11-00661-t001:** Description of co-morbidities and medication use in the study sample.

Socio-Demographic and Clinical Characteristics	N (%) or Mean (SD)
Nº of co-morbidities	5.7 (3.1)
Educational level:	
no studies	125 (38.8%)
primary studies	136 (42.2%)
>primary studies	61 (19.0%)
Arthritis	169 (52.5%)
Ischaemic heart disease	69 (21.4%)
Peripheral arterial disease	50 (15.5%)
Stroke	32 (9.9%)
Depression	63 (19.6%)
Chronic bronchitis	44 (13.7%)
Asthma	26 (8.1%)
Diabetes	78 (78.2%)
Chronic liver disease	8 (2.5%)
Prostatic syndrome (men)	66 (39.3%)
Hypertension	225 (70.1%)
Dyslipidaemia	157 (50.8%)
Glomerular filtration < 60	80 (24.8%)
BMI	28.8 (3.6)
Obesity (BMI ≥ 30)	101 (31.2%)
Malnutrition/malnutrition risk (MNA_sf < 12)	17 (5.4%)
Fiber intake ≤ 21 g/day	86 (26.5%)
Insulin resistance	81 (25.4%)
Nº of medications	5 (3.1)
Oral corticosteroids	5 (1.5%)
Oral antidiabetics	68 (21.2%)
Benzodiazepines	102 (31.5%)
Antipsychotics	1 (0.3%)
NSAIDs or paracetamol	181 (55.9%)
Diuretics	124 (38.3%)
ACEIs	93 (28.7%)
ARBs	76 (23.5%)
Beta-blockers	57 (17.6%)
PPIs	163 (50.8%)
Statins	155 (47.8%)
SSRIs	44 (13.6%)
Antiepileptics	23 (7.1%)

ICW/LM ratio as the dependent variable. ACEI, angiotensin-converting enzyme inhibitor; ARB, angiotensin II receptor blocker; BMI, body mass index; NSAID, non-steroidal anti-inflammatory drug; PPI, proton pump inhibitor; SD, standard deviation; SSRI, selective serotonin reuptake inhibitor.

**Table 2 nutrients-11-00661-t002:** Relationship between ICW/LM ratio (in mL/kg) and indicators of muscle strength and functional capacity (continuous variables).

Indicators of Functional Capacity	r_s_	*p*	Β *	*p*
Hand grip (kg)	0.397	<0.001	0.117	<0.001
Barthel score	0.317	<0.001	0.059	<0.001
Gait speed (m/s)	0.311	<0.001	0.003	<0.001
TUG (s)	−0.326	<0.001	−0.031	<0.001
Outdoor walking (h/day)	0.268	<0.001	0.283	<0.001

* ICW/LM ratio as the independent variable. ICW, intracellular water; LM, lean mass; TUG, timed up-and-go. r_s_, Spearman correlation coefficient; β, linear regression coefficient.

**Table 3 nutrients-11-00661-t003:** Relationship between ICW/LM ratio (in mL/kg) and indicators of functional capacity and frailty (categorical variables).

Indicators of Functional Capacity	Mean; SD (*N*) ICW/LM Ratio When Condition Present	Mean; SD (*N*) ICW/LM Ratio When Condition not Present	*p*	Effect Size
Frailty	391.0; 26.1 (46)	411.1; 28.8 (278)	<0.001	−0.70
Weight loss	377.1; 21.7 (16)	409.9; 28.7 (308)	<0.001	−1.14
Exhaustion	396.7;32.5 (64)	411.1; 27.7 (260)	<0.001	−0.52
Poor muscle strength	398.6; 32.0 (104)	412.8; 26.8 (220)	<0.001	−0.53
Poor gait speed	391.3; 27.6 (67)	412.7; 28.1 (257)	<0.001	−0.76
Poor physical activity	399.4; 32.2 (100)	412.2; 27.0 (224)	<0.001	−0.47
Outdoor life	409.4; 28.7 (288)	398.9; 33.0 (35)	0.046	0.32
Unable to stand on 1 foot for 5 s	411.9; 29.6 (233)	399.0; 26.4 (91)	<0.001	0.49
Poor physical activity	397.5; 32.8 (81)	411.9; 27.1 (243)	<0.001	−0.53

ICW, intracellular water; LM, lean mass; SD, standard deviation; effect size, (mean of the group with the condition–mean of the group without the condition)/standard deviation of the group with the condition.

**Table 4 nutrients-11-00661-t004:** Multivariate analysis results for the independent effect of ICW/LM ratio (in mL/kg) on muscle strength, functional capacity and frailty.

Independent Variables in the Model	Muscle Strength (Hand Grip in kg)		Barthel Score		Frailty	
	β (95% CI)	*p*	β (95% CI)	*p*	OR (95% CI)	*p*
ICW/LM ratio (mL/kg)	0.027 (0.01; 0.05)	0.007	0.031 (0.01; 0.05)	0.007	0.98 (0.97; 0.99)	0.011
Age (year)	−0.121 (−0.28; 0.04)	0.146	−0.183 (−0.37; −0.001)	0.048	1.03 (0.92; 1.14)	0.612
Female sex	−10.92 (−12.8; −9.08)	<0.001	−2.766 (−4.81; −0.72)	0.008	7.15 (1.83; 27.9)	0.005
Nº of comorbidities	−0.857 (−1.15; −0.57)	<0.001	−0.992 (−1.32; −0.67)	<0.001	1.74 (1.42; 2.13)	<0.001
Lean mass (Kg)	0.225 (0.12; 0.33)	<0.001	−0.019 (−0.14; 0.10)	0.755	1.04 (0.96; 1.12)	0.296

ICW, intracellular water; LM, lean mass; OR, odds ratio; TUG, timed up-and-go test.

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
