# Peer review of "Intracellular Water Content in Lean Mass is Associated with Muscle Strength, Functional Capacity, and Frailty in Community-Dwelling Elderly Individuals. A Cross-Sectional Study"

_nutrients, 2019, doi:10.3390/nu11030661_

Round 1
Reviewer 1 Report
INTRODUCTION
I think the introduction is well structured and write. As a contribution, mention that in the second paragraph where we talk about muscle, perhaps we could mention the importance of this tissue in the elderly health.
METHODS
I think that the basis of the BIAS methodology is largely explained but other things such as the protocol used in the hand-grip are not specify... (everyone has done it with the same arm? or has a protocol been followed where the test is done with the dominant arm? how many repetitions per person?)
Why is the analysis of dichotomous variables not performed with the ICW/LM with an ANCOVA test?
RESULTS
I consider that the results shown in this study should be stated more clearly. It is difficult to identify which data are really important.
The descriptive and frequency results of the analyzed variables are not shown. I think that taking into account the sample (n = 324) and the number of comorbidity variables (a total of 14) and chronic medications (13) would be important to consider. Because some of the results could be influenced by being obtained with a small sample number.
In my opinión tables contains so many stripes and are not clear enough. It would be a good idea to removed some of them and used bold or italics in each colum name (p, rs, presence/yes...)
In Table 2 muscle strenght, barthel score and frailty are analyzed with the ame variables. It would be a good option put it horizontally. Not to be repetitive. Moreover, in the Frailty parameters why it is shown OR instead of beta value?
DISCUSSION
I consider that the discussion should focus more on the relevance of the results in clinical practice (the importance of identifying the fragility status from ICW)
I think it is not very correct to discuss the results of a cross sectional study in elderly individuals with those obtained in studies of young athletes (reference 21) or rats (reference 24).
In my opinion it is important to highlight the results obtained. Extrapolate them to a practical utility on a day-to-day basis. Perhaps the results on ICW when associated with muscle strength, functional capacity and fragility could be an indicator of the functional health of elderly individuals...
Author Response
Answers to reviewer 1:
Thank you very much for your comments and suggestions.
1. I think the introduction is well structured and write. As a contribution, mention that in the second paragraph where we talk about muscle, perhaps we could mention the importance of this tissue in the elderly health.
Answer: As the reviewer suggest, we have introduced a new sentence in the second paragraph of the introduction regarding the importance of muscle in elderly health and successful ageing (line 56-57). We have also introduced a new reference to support this sentence.
2. I think that the basis of the BIAS methodology is largely explained but other things such as the protocol used in the hand-grip are not specify... (everyone has done it with the same arm? or has a protocol been followed where the test is done with the dominant arm? how many repetitions per person?
Answer: According to the reviewer comment, we have added in the methodology section a more detailed explanation of hand–grip measurement. We provide a reference with more extensive explanation of the study protocol and methodology.
3. Why is the analysis of dichotomous variables not performed with the ICW/LM with an ANCOVA test?
Answer: Regarding the statistical analysis to compare the ICW/LM ratio between two groups, we used the Student t test for independent variables or the Mann Whitney U test, depending on the normal distribution of the ICW/LM ratio in the analyzed groups. We aimed to explore the crude relationship between these variables and were not interested in the adjusted association, or in causal relationships.
4. I consider that the results shown in this study should be stated more clearly. It is difficult to identify which data are really important.
Answer: The results section is organized in three paragraphs. The first one describes the ICW/LM ratio for the overall sample and by sex groups. The second one presents the crude and adjusted relationships between ICW/LM ratio and functional capacity indicators and frailty, which may be possible consequences of cell dehydration, and are the main results of the study. Finally, the third paragraph presents the association of ICW/LM ratio with other study variables, which could be causes or determinants of low ICW/LM ratio. In order to provide a more clearly presentation of results, we have introduced subheadings in the results section.
5. The descriptive and frequency results of the analyzed variables are not shown. I think that taking into account the sample (n = 324) and the number of comorbidity variables (a total of 14) and chronic medications (13) would be important to consider. Because some of the results could be influenced by being obtained with a small sample number.
Answer: As the reviewer suggest, we have introduced a new table with the description and frequencies of the co-morbidities and medications in the study sample. These data allow better interpreting the study results.
6. In my opinion tables contains so many stripes and are not clear enough. It would be a good idea to removed some of them and used bold or italics in each colum name (p, rs, presence/yes...)
Answer: To make tables more clear we have divided table 1 in to two tables, and have introduced some small changes: new headings in some columns, bold in statistically significant results ...
7. In Table 2 muscle strenght, barthel score and frailty are analyzed with the ame variables. It would be a good option put it horizontally. Not to be repetitive. Moreover, in the Frailty parameters why it is shown OR instead of beta value?
Answer: As the reviewer suggest we have remade the original table 2 in a horizontal shape. Frailty results are expressed in terms of OR because Frailty is a dichotomous variable (not a continuous one), so in this case we have used logistic regression (no lineal regression).
8. I consider that the discussion should focus more on the relevance of the results in clinical practice (the importance of identifying the fragility status from ICW).
Answer: in response to the reviewer comment, we have added a sentence in the discussion regarding the potential impact of the results in clinical practice.
9. I think it is not very correct to discuss the results of a cross sectional study in elderly individuals with those obtained in studies of young athletes (reference 21) or rats (reference 24).
Answer: I’m sorry, but I’m not sure to correctly understand this comment. With these two references we are not comparing our results with those of other studies focused on young athletes and in rats. But these two references help us to support and reinforce the idea that: a) ICW is related with muscle strength, also in other populations such as young athletes (we corroborate in the elderly this relationship previously observed in the young), and b) that dehydration affects mainly skeletal muscle tissue (which, because ethical limitations, is difficult to evaluate in humans). These are two main ideas in our discussion and in the interpretation of our results. It is largely accepted that clinical experimentation must be based on pre-clinical studies and, although the necessary prudence in the extrapolation of animal experimentation results in human beings, these results may be of great interest and can serve as the rationale for clinical investigation. We are the opinion that these two references are relevant and pertinent in the discussion section.
10. In my opinion it is important to highlight the results obtained. Extrapolate them to a practical utility on a day-to-day basis. Perhaps the results on ICW when associated with muscle strength, functional capacity and fragility could be an indicator of the functional health of elderly individuals...
Answer: As mentioned in point 8, we have added a sentence in the discussion regarding the potential utility of the results in clinical practice.

Reviewer 2 Report
This is an interesting study examining the relationship between ICW/LM ratio and function in older adults. High ICW/LM ratio seemed to be related to better physical function.
Some comments need to be addressed:
Introduction:
Line 58 please add a reference after «fat mass»
Table 1: What is the «outdoor life» variable? What is the difference with «Outdoor walking»? Even though the methods are published elsewhere, this could help the readers understand this outcome.
Line 76: Regarding the methods published, it took me 15 minutes to find the correct article. Please remove the two citations and use this reference:
M. Serra-Prat et al. / European Geriatric Medicine 7 (2016) 531–537
The statistical analysis section needs to be improved. For each analyses, what were the confounders? And did the authors adjusted for medication consumption, food/beverage intake and BMI? Since TBW can be influenced by a multitude of variables, it may be of interest to display an adjusted and non/adjusted model.
Did the authors check for statistical power and for multiplicity? Given that there is a lot of analyses, the change of detecting something significant is higher. What program did the authors used for performing the statistical analyses?
Table 1 is difficult to understand. I think its better to separate the Spearman's correlation and the Mann-Whitney test for better understanding. Also, what is the meaning of the direction of the effect size? What does a negative effect size mean? Please add legends to these tables.
No data between, robust, frail and prefrail are presented, which could be of interest.
Discussion
Main comment: the authors should compare their findings with this study published in JSCR in 2018:
Impact of Mild hypohydration on muscle endurance, power, and strength in healthy, active older men Goulet et al, 2018 Journal of Strength and Conditioning research.
The first paragraph needs to be written, the sentence is too long.
Line 185: this sentence has too many «and».
Author Response
Answers to reviewer 2:
Thank you very much for your comments and suggestions.
1. Line 58 please add a reference after «fat mass»
Answer: As the reviewer suggest, we have added a new reference to justify the relative increase of fat mass with age.
2. Table 1: What is the «outdoor life» variable? What is the difference with «Outdoor walking»? Even though the methods are published elsewhere, this could help the readers understand this outcome.
Answer: Following the reviewer recommendation, we have added in the methods section a specification of what “outdoor life” and “outdoor walking” meant.
3. Line 76: Regarding the methods published, it took me 15 minutes to find the correct article. Please remove the two citations and use this reference: M. Serra-Prat et al. / European Geriatric Medicine 7 (2016) 531–537
Answer: In response to this comment, we have removed the two mentioned citations and changed by the new one suggested by the reviewer.
4. The statistical analysis section needs to be improved. For each analyses, what were the confounders? And did the authors adjusted for medication consumption, food/beverage intake and BMI? Since TBW can be influenced by a multitude of variables, it may be of interest to display an adjusted and non/adjusted model.
Answer: Thank you for this comment. We have modified the text in the statistical analysis section to improve its comprehension. As mentioned, multiple regression analyses included in the model those variables that showed a significant relationship with the dependent variable in the bivariate analysis. We have specified in the text which are these variables (possible confounders), and stated that medication was not included because of multi-collinearity with number of co-morbidities. Food/beverage intake was not analyzed in the whole study and BMI was not included in the multivariate analysis because did not show any association with the dependent variables (and so, could not act as a confounder). Adjusted effects are presented in table 2 and unadjusted effects in tables 1 and 3.
5. Did the authors check for statistical power and for multiplicity? Given that there is a lot of analyses, the change of detecting something significant is higher. What program did the authors used for performing the statistical analyses?
Answer: the reviewer is right indicating that when several statistical tests are performed, the probability of identifying at least one significant result due to chance increases. The more hypothesis tested the higher chance for a false positive result (type I error). The Bonferroni correction addresses this question. However, it is not free from controversy because: a) it does not identify in which test result there is the error, b) as each comparison is independent from the others, the probability of a type I error is also independent in each comparison, or c) when decreasing the probability of a type 1 error, the probability of a type 2 error (false negative result) increases. We understand that the results obtained must be interpreted not only according to the probability of a type 1 error (p value), but also according to the magnitude of the observed effect, its robustness and its clinical and biological plausibility. On the other hand, as the reviewer suggest, we have added in the text the statistical program used for the analysis (SPSS 15.0).
6. Table 1 is difficult to understand. I think its better to separate the Spearman's correlation and the Mann-Whitney test for better understanding. Also, what is the meaning of the direction of the effect size? What does a negative effect size mean? Please add legends to these tables.
Answer: As mentioned before, to make tables more clear we have introduced some changes such as new headings in some columns. As the reviewer suggest we have separate the original table 1 in two different tables (one for numerical variables and the other for categorical ones). In the table legend and for a better understanding, we have specified the meaning of rs (Spearman correlation coefficient), β (linear regression coefficient), and the definition of effect size. A negative effect size indicates that the mean of the group without the condition (unexposed) is under the mean of the group with the condition (exposed group).
7. No data between, robust, frail and prefrail are presented, which could be of interest.
Answer: As the reviewer suggest, we have added in the results section data on the ICW/LM ratio for the 3 groups of frailty status (robust, pre-frail and frail).
8. Main comment: the authors should compare their findings with this study published in JSCR in 2018: Impact of Mild hypohydration on muscle endurance, power, and strength in healthy, active older men Goulet et al, 2018 Journal of Strength and Conditioning research.
Answer: We have introduced in the discussion a comment comparing our results with those of Goulet et al 2018. We have also added this references in the bibliography.
9. The first paragraph needs to be written, the sentence is too long. Line 185: this sentence has too many «and».
Answer: In response to the reviewer suggestion we have rewritten the first paragraph of the discussion, and one “and” in the indicated sentence has been removed.

Round 2
Reviewer 1 Report
I think the authors have improved the writing of the article. The results they show are better understood than in the previous version. In my opinion, it would be important to maintain the same format in the tables and forms of the text but I consider some changes of format and the revision of the enumeration of the referencesshould be made.
Table 4 is not included same format as above. I consider that maintaining the same format throughout the article will facilitate the comprehension of the work and make reading easier when it comes to revision.
References numbers might be revised. The reference corresponds to Goulet et al in the text is marked as number 21. However, in the references the number 21 refers to Yamada et al. and Goulet is in the number 23...
Author Response
Answers to reviewer 1.
· I think the authors have improved the writing of the article. The results they show are better understood than in the previous version.
Answer: Thank you for this comment.
· Table 4 is not included same format as above. I consider that maintaining the same format throughout the article will facilitate the comprehension of the work and make reading easier when it comes to revision.
Answer: I think that in the last version of the manuscript (nutrients-445203-revised1.docx), all tables are in the same format (maybe there is a problem with versions). What is the reviewer referring when talking about “the same format”?
· References numbers might be revised. The reference corresponds to Goulet et al in the text is marked as number 21. However, in the references the number 21 refers to Yamada et al. and Goulet is in the number 23...
Answer: As the reviewer has observed, there was a mistake in the number of the reference by Goulet et al. We have corrected this error in the text (is reference 23) and we have also checked the rest of references.

Reviewer 2 Report
I still feel that there is a lot of work before accepting this paper. Here below are my comments. Mainly, two sections should be removed (discussed below). These two sections (results on medications and the discussion of age) is put in front, but the objectives of this study was not focused on these outcomes. As such, this article loses focus and and needs to be more structured.
Methods section
Line 92: please indicate specifically on how the authors ensured dehydration.
Hydration status: please indicate if they were any protocols on hydration status. Did the participants had specific instructions as to where they had to drink water before the experiment? Were they refrained from taking diuretics before the visits? How about hematocrit levels? I think this may be an important limit of this study because ECW can vary enormously, especially in older adults.
Line 113, add strength after hand grip.
Results
Table 1 is lacking of baseline characteristics data of the sample: Age, education, sex, BMI.
Authors should run the same analyses using the same outcomes but with lean mass (LM) instead of ICW/LM ratio and insert them in the supplementary materials. This sensitivity analyses will help answer the problem indicated in the introduction (line 65).
Table 3: can the authors indicate the sample of people with/without the condition?
Table 4: The authors state in the discussion a relationship between ICW/LM ratio and frailty. But in table 4, the OR between ICW/LM is 0.98, which seems to be very low. The authors should discuss this accordingly. Is it possible to give the 95% CI and include LM?
Finally, the make the paper more straightforward, the authors should consider removing the analysis on clinical conditions and medications. This was not part of the objective and it is not mentioned in the title.
Discussion
In general, the discussion still needs some work. It is difficult to see the links between water content and muscle. Furthermore, the title of this study, which is related to muscle strength and frailty does not reflect the focus of the discussion. For example, the main finding based on the authors is that ICW/LM ratio decreases with age, which cannot be confirmed with this study, because it was measured at baseline. Caution must be taken in appropriately reporting the results.
In line with this, the discussion focuses on this relationship with age, which was not the original objective of this study. The authors should completely remove this section, because it was not the point of the study and also it hides other important findings such as muscle strength, function and frailty.
Line 188-193: I am not sure that having cross-sectional results from a small sample size can be a plausible, coherent and consistent indicator of muscle mass and function. I am not sure about this statement. More studies are needed to confirm that ICW/LM ratio can be used in a clinical setting. The authors should remove this paragraph or tone down the expectations. Also, even though they have lower ratio score, we do not know if this score is clinically relevant. This can be discussed in the limitations.
I think there are links missing between water content, kidney function and muscle mass. Could the authors elaborate on this? There are many studies showing prospective relationships between impaired kidney function and muscle mass. The authors should redo a search on the litterature regarding the mechanisms of sarcopenia and what does ICW/LM is related to.
The discussion is still laking on how higher water content can influence muscle strength (muscle contraction). What are the potential mechanisms that can explain this association? Some theories exists on dehydration and increased risk for cramps. I am wondering if ICW/LM could influence neuromuscular function? How about Ca2+ release, acetylcholine receptors affinity to acetylcholine at the junction? In any case, the authors should also take the time to read the literature regarding water content, and muscle contraction.
Line 233-234, these mechanisms are more associated to muscle mass than muscle function. I recommend the authors read the seminal work from Clarke & Manini : https://www.ncbi.nlm.nih.gov/pmc/articles/PMC3260480/
Finally, since this is a cross-sectional study, the fact that there is a relationship with ICW/LM and strength may be the fact that LM is lower than those with having worse function scores. Hence, we do not know the direction of the association. The authors should discuss this in the limitations section.
Author Response
Answers to Reviewer 2 (second round).
· I still feel that there is a lot of work before accepting this paper. Here below are my comments. Mainly, two sections should be removed (discussed below). These two sections (results on medications and the discussion of age) is put in front, but the objectives of this study was not focused on these outcomes. As such, this article loses focus and and needs to be more structured.
Answer: Thank you for this comment and for the opportunity to improve the manuscript quality, focus and comprehension. It is true that the focus of the study is on the relationship of the ICW/LM ratio with muscle strength, functional capacity and frailty. The objective of the study also mentioned the association with “other clinical characteristics” but, according to the reviewer comment, in order to improve focus and clarity, we have removed from all the manuscript (results and discussion) data about co-morbidities and medications. We believe that data about age is of interest and can help interpreting the overall results, so we have not remove this part but we have made an effort to synthesize it.
· Line 92: please indicate specifically on how the authors ensured dehydration.
Answer: As the reviewer suggests, we have explained in more detail how dehydration was clinically assessed before BIA analysis and the clinical signs and symptoms evaluated.
· Hydration status: please indicate if they were any protocols on hydration status. Did the participants had specific instructions as to where they had to drink water before the experiment? Were they refrained from taking diuretics before the visits? How about hematocrit levels? I think this may be an important limit of this study because ECW can vary enormously, especially in older adults.
Answer: The reviewer is right indicating that hydration status is a relevant aspect. We have some instructions for the correct realization of BIA, which include no intense physical exercise 24h before the test, no alcohol consumption 8 h before the test, no water or fluid intake 2 h before the test or go to the toiled just before the test. Patients received these instructions when they were cited to the assessment visit. However, we cannot ensure full compliance of these instructions by the patient. That’s why we have added a sentence in this regard in the limitation section of the discussion.
· Line 113, add strength after hand grip.
Answer: as the reviewer suggest we have added the word “strength” after hand grip.
· Table 1 is lacking of baseline characteristics data of the sample: Age, education, sex, BMI.
Answer: description of age and sex variables of the study sample are presented in the text (first line of results section). In order not to repeat information, these data were not included in table 1. As the reviewer suggest, we have added education and BMI description in table 1.
· Authors should run the same analyses using the same outcomes but with lean mass (LM) instead of ICW/LM ratio and insert them in the supplementary materials. This sensitivity analyses will help answer the problem indicated in the introduction (line 65).
Answer: According to the reviewer suggestion, we have run the same analyses using the same outcomes but with lean mass (LM) instead of ICW/LM ratio and we provide these results in the supplementary materials. Moreover, in order to differentiate the effect of cell hydration (ICW/LM ratio) from the effect of LM (which include muscle mass) on muscle strength, functional capacity and frailty, we have included LN as an independent variable in the multivariate models (see updated table 4). ICW/LM ratio continues to show a statistical significant effect after adjusting for LM.
· Table 3: can the authors indicate the sample of people with/without the condition?
Answer: As the reviewer indicates, we have added two columns in table 3 with the number of patients with or without the condition.
· Table 4: The authors state in the discussion a relationship between ICW/LM ratio and frailty. But in table 4, the OR between ICW/LM is 0.98, which seems to be very low. The authors should discuss this accordingly. Is it possible to give the 95% CI and include LM?
Answer: We state that there is a relationship between ICW/LM ratio and frailty because, as shown in table 4, this relationship showed an adjusted OR with a p=0.011. This statistically significant effect indicates a 2% reduction in the risk of frailty for each mL of ICW/kg LM (independently of age, sex, nº of co-morbidities and LM). We do not believe that this is a very low effect, as an increase of 25 mL of ICW/kg LM (approximately 1 standard deviation) would correspond to a 50% decrease in the risk of frailty. In response to the reviewer comment, we have added a sentence in the discussion to interpret this independent relationship. We have added in table 4 the 95% CI. We have repeated the multivariate analysis including LM in the multivariate models, and the adjusted effect of ICW/LM ratio does not vary and remains statistically significant (see updated table 4). Results on the effect LM on main outcome measures are available in the supplementary material.
· Finally, the make the paper more straightforward, the authors should consider removing the analysis on clinical conditions and medications. This was not part of the objective and it is not mentioned in the title.
Answer: As the reviewer suggest and in order to improve the focus of the manuscript we have removed the analysis of clinical conditions and medications.
· In general, the discussion still needs some work. It is difficult to see the links between water content and muscle. Furthermore, the title of this study, which is related to muscle strength and frailty does not reflect the focus of the discussion. For example, the main finding based on the authors is that ICW/LM ratio decreases with age, which cannot be confirmed with this study, because it was measured at baseline. Caution must be taken in appropriately reporting the results. In line with this, the discussion focuses on this relationship with age, which was not the original objective of this study. The authors should completely remove this section, because it was not the point of the study and also it hides other important findings such as muscle strength, function and frailty.
Answer: As the reviewer suggest we have tried to focus the discussion on the relationship between water and muscle function. However, it is difficult not referring to age, sex and co-morbidity effects on ICW/LNM ratio because they are related with both the body hydration and muscle strength. So, when assessing the independent effect of ICW/LM ratio on strength it is of interest to adjust for these other variables. The reviewer is also right indicating that with the cross-sectional design the effect of age cannot be confirmed but suggested. We mention it in the limitation section. Again, we have revised all the discussion to put the emphasis on the water-muscle strength relationship.
· Line 188-193: I am not sure that having cross-sectional results from a small sample size can be a plausible, coherent and consistent indicator of muscle mass and function. I am not sure about this statement. More studies are needed to confirm that ICW/LM ratio can be used in a clinical setting. The authors should remove this paragraph or tone down the expectations. Also, even though they have lower ratio score, we do not know if this score is clinically relevant. This can be discussed in the limitations.
Answer: We probably express ourselves badly in this sentence. We have rewritten this entire paragraph in order to tone down the possible relevance of this indicator in clinical investigation.
· I think there are links missing between water content, kidney function and muscle mass. Could the authors elaborate on this? There are many studies showing prospective relationships between impaired kidney function and muscle mass. The authors should redo a search on the litterature regarding the mechanisms of sarcopenia and what does ICW/LM is related to.
Answer: As the reviewer suggest, we have introduced in the discussion the following paragraph regarding the relationship between renal function and muscle wasting.
“It’s known that chronic kidney disease (CKD) results in muscle wasting (Gollie 18). CKD favors protein degradation because the activation of the ubiquitin–proteasome system, which can be initiated by complications of CKD, such as metabolic acidosis, defective insulin signaling, inflammation or impaired micro-RNA responses (Wang 14). CKD is also related with impaired water homeostasis and hyperosmotic stress. Despite this, the mechanisms by which hyperosmotic stress and cell dehydration affects muscle strength have been little studied. It is unknown to what extent the alteration of muscle function is due to a direct effect of cellular dehydration on actin-myosin interaction as suggested by Grazi (Grazi) or is due to a process of protein degradation related with inflammation and favored by alterations in renal function, so further research is required in this regard.”
· The discussion is still laking on how higher water content can influence muscle strength (muscle contraction). What are the potential mechanisms that can explain this association? Some theories exists on dehydration and increased risk for cramps. I am wondering if ICW/LM could influence neuromuscular function? How about Ca2+ release, acetylcholine receptors affinity to acetylcholine at the junction? In any case, the authors should also take the time to read the literature regarding water content, and muscle contraction.
Answer: As the reviewer indicate, we have searched about the link between water and muscle function and added in the discussion the following paragraph. It helps focusing the discussion on the relation between ICW and muscle function. Two new references by Grazi have been introduced.
“Grazi has studied the interaction between water and muscle contractile proteins. The muscle cell contains thousands of myofibrils, formed by thin filaments (actin) and thick filaments (myosin), which interact to produce the shortening or contraction of the muscle. According to Grazi, the participation of water in muscle contraction is due to the hydrophilic nature of the proteins of the contractile apparatus. This property implies that a small change in the concentration of these proteins induces a significant change in the osmotic pressure, as well as in the chemical potential of these proteins, which is related to the stiffness of the contractile structure (ability to oppose distension). Additionally, in vitro studies show that when the proteins of the contractile apparatus are well hydrated they assemble to form ordered structures. These authors describe that in the actin-myosin binding, the osmotic pressure determines both the distance between filaments and the elastic force that acts on the structure of each protein, so that this system links the osmotic pressure with the elastic reaction of the components of the cross-bridge (). Overall, these studies have related water with muscle contraction and indicate that water determines sarcomere length, sarcomere stretching and cross-bridge attachment and detachment (Grazi 06, 08).”
· Line 233-234, these mechanisms are more associated to muscle mass than muscle function. I recommend the authors read the seminal work from Clarke & Manini : https://www.ncbi.nlm.nih.gov/pmc/articles/PMC3260480/
Answer: thank you very much for this reference. We have read it carefully and introduced the concept of dynapenia in the discussion. Two new references by Clark and Manini have also been introduced.
“Age-related loss of muscle strength is only partially explained by age-related loss of muscle mass (). That’s why Manini et al proposed the term dynapenia to describe age-related loss of muscle strength and power and differentiate it from the sarcopenia concept, which, in the opinion of these authors, must be restricted to age-related loss of muscle mass (). Other factors related with muscle quality instead of muscle quantity must explain muscle weakness in elderly subjects. In our opinion, among these factors, myocite hydration could play a relevant role.”
· Finally, since this is a cross-sectional study, the fact that there is a relationship with ICW/LM and strength may be the fact that LM is lower than those with having worse function scores. Hence, we do not know the direction of the association. The authors should discuss this in the limitations section.
Answer: The ICW/LM ratio indicates the quantity of ICW per 1kg of LN, so it allows camparing individuals with different LM. As mentioned in the limitation section, the cross-sectional design does not allow to establish the directionality of relationships or to establish causal relationships.
